# A Hydraulic Pump Fault Diagnosis Method Based on the Modified Ensemble Empirical Mode Decomposition and Wavelet Kernel Extreme Learning Machine Methods

**DOI:** 10.3390/s21082599

**Published:** 2021-04-07

**Authors:** Zhenbao Li, Wanlu Jiang, Sheng Zhang, Yu Sun, Shuqing Zhang

**Affiliations:** 1Hebei Provincial Key Laboratory of Heavy Machinery Fluid Power Transmission and Control, Yanshan University, Qinhuangdao 066004, China; lizhenbao@stumail.ysu.edu.cn (Z.L.); zsqhd@ysu.edu.cn (S.Z.); zhang_py@stumail.ysu.edu.cn (Y.S.); 2Key Laboratory of Advanced Forging & Stamping Technology and Science, Yanshan University, Ministry of Education of China, Qinhuangdao 066004, China; 3School of Electrical Engineering, Yanshan University, Qinhuangdao 066004, China; zhshq-yd@163.com

**Keywords:** hydraulic pump, fault diagnosis, modified ensemble empirical mode decomposition (MEEMD), wavelet kernel extreme learning machine (WKELM)

## Abstract

To address the problem that the faults in axial piston pumps are complex and difficult to effectively diagnose, an integrated hydraulic pump fault diagnosis method based on the modified ensemble empirical mode decomposition (MEEMD), autoregressive (AR) spectrum energy, and wavelet kernel extreme learning machine (WKELM) methods is presented in this paper. First, the non-linear and non-stationary hydraulic pump vibration signals are decomposed into several intrinsic mode function (IMF) components by the MEEMD method. Next, AR spectrum analysis is performed for each IMF component, in order to extract the AR spectrum energy of each component as fault characteristics. Then, a hydraulic pump fault diagnosis model based on WKELM is built, in order to extract the features and diagnose faults of hydraulic pump vibration signals, for which the recognition accuracy reached 100%. Finally, the fault diagnosis effect of the hydraulic pump fault diagnosis method proposed in this paper is compared with BP neural network, support vector machine (SVM), and extreme learning machine (ELM) methods. The hydraulic pump fault diagnosis method presented in this paper can diagnose faults of single slipper wear, single slipper loosing and center spring wear type with 100% accuracy, and the fault diagnosis time is only 0.002 s. The results demonstrate that the integrated hydraulic pump fault diagnosis method based on MEEMD, AR spectrum, and WKELM methods has higher fault recognition accuracy and faster speed than existing alternatives.

## 1. Introduction

In recent years, modern industrial equipment has developed in the direction of complexity, high speed, and intelligence. This not only ensures the quality of the products, but also greatly reduces the associated production costs and significantly improves production efficiency. However, in the actual production process, due to the continuous expansion of system scale and the increasing complexity of the structures involved, the probability of equipment failure has been gradually increasing. Therefore, it is very important to improve the safety and reliability of such equipment. In this regard, it is necessary not only to improve the reliability of the equipment in the process of equipment design and manufacturing, but also to carry out real-time status monitoring and fault diagnosis during the operation of the equipment. In this way, we can detect the fault initially, then predict the location and cause of the fault, in order to provide accurate information for predictive maintenance. Therefore, more and more attention has been paid to fault diagnosis and prediction technology, for which a lot of research work has been carried out [1,2,3,4].

Hydraulic systems have the advantages of small size, light weight, high power, smooth transmission, and fast response. Hydraulic systems have been widely used in the aerospace, shipbuilding, metallurgical, machine tool, agricultural machinery, and engineering machinery industries, among other fields. The wide application of hydraulic systems has greatly promoted the development of industrial production [5]. Since entering the 21st century, hydraulic systems and hydraulic equipment have become more complicated and large-scale. Once a failure occurs, it not only can damage the equipment, but may even cause casualties [6]. Hydraulic pumps have the characteristics of complex structure, high speed, high pressure, and continuous operations, and are the most prone to failure. Therefore, fault diagnosis technology for hydraulic systems has become a research hotspot in recent years.

In fault diagnosis, the first step is to extract fault features from the acquired data [7,8]. Feature extraction is the process of obtaining fault features through signal processing [9,10]. Considering that the vibration signal of a hydraulic pump has non-linear and non-stationary characteristics, it is difficult to extract the fault features accurately by using traditional time- and frequency-domain analyses. The key information of the hydraulic pump vibration signal can be extracted more accurately by using time-frequency characteristic analysis methods. Time-frequency analysis methods include non-adaptive analysis and adaptive analysis. Adaptive analysis refers to the modal decomposition and reconstruction of the signal, according to the characteristics of the signal, in the process of signal processing, in order to achieve good analysis results.

Considering the mode mixing phenomenon of the empirical mode decomposition (EMD) method [11,12], Wu et al. proposed the ensemble empirical mode decomposition (EEMD) method, which effectively suppresses modal mixing by adding Gaussian white noise to the original signal [13]. In view of the shortcomings of the EEMD method, scholars have carried out a lot of research. Finally, Zheng et al. proposed the modified ensemble empirical mode decomposition (MEEMD) method [14], which adds a pair of Gaussian white noises with the same amplitude and opposite signs, such that they cancel each other and the influence of the noise on the signal is reduced. Compared with the EEMD method, this method has a better decomposition effect. Jiang et al. applied the MEEMD method to the fault diagnosis of the main shaft bearing of a wind turbine [15]. Their method showed good results, in terms of training accuracy and training speed, which verifies the feasibility of this method. Zheng proposed a method combining MEEMD and generalized adaptive S transform (AGST), which was successfully applied to the vibration signal analysis of an internal combustion engine [16]. Shi et al. applied the MEEMD method to fault feature extraction of rolling bearings, which not only effectively suppressed the phenomenon of modal mixing but also greatly reduced the number of pseudo-components and accurately extracted the weak fault information of rolling bearings [17]. Although the MEEMD method has been widely used in the field of fault diagnosis, it has seldom been reported in hydraulic system fault diagnosis. Therefore, in this paper, we applied it to the vibration signal feature extraction of an axial piston pump, in order to improve the accuracy of fault identification.

In 2004, Professor Huang from Nanyang Technological University in Singapore proposed a new single-hidden layer feedforward neural network, called the extreme learning machine (ELM) [18,19,20]. In 2018, Zhang et al. proposed a method combining evidence theory and ELM, which effectively solved the classification problem of incomplete data [21]. However, it should be noted that the optimal number of neurons in the hidden layer obtained by the incremental extreme learning machine and its improved method may be very large, thus wasting a lot of unnecessary training time.

As the ELM method needs to set the number of hidden layer neurons and randomly select input weights and hidden layer biases, ELM instability is inevitable. Therefore, Huang et al. introduced a kernel function into the ELM to propose the kernel-ELM (KELM) method. This method avoids the problems of artificially setting the number of hidden layer neurons and randomly selecting the input weights and hidden layer biases in ELM by means of kernel function mapping. KELM can achieve very high training accuracy by using the powerful mapping ability of kernel function [22]. To date, KELM has been widely used in various fields. One study [23] has proposed a parallel KELM, which effectively improved the computing speed of KELM. Cho et al. applied a K-ELM to correctly classify the stress levels to reflect five different levels of stress situations [24]. Zhang et al. proposed a soft sensing algorithm that can detect alumina concentration by the electrical signals such as voltages and currents of the anode rods [25]. Yang et al. combined the multi-scale feature energy with the KELM to effectively diagnose the airflow jamming fault of a fluidized bed [26]. Zhang et al. applied the KELM to the fault diagnosis of a coal mill and optimized the kernel parameters with particle swarm optimization (PSO) [27]. Yang et al. proposed a deep kernel extreme learning machine (DK-ELM), which has been successfully applied to aero-engine fault diagnosis [28]. Considering that wavelet functions have the advantages of time–frequency local analysis and multi-resolution analysis, a wavelet kernel function has been introduced into KELM to construct the wavelet kernel extreme learning machine (WKELM).

In this paper, a fault diagnosis method for a hydraulic pump based on the integration of the MEEMD, autoregressive (AR) spectrum energy, and WKELM methods is proposed and studied. First, the vibration signals of an axial piston pump under different working states were acquisited from a hydraulic pump fault simulation test bed. Secondly, signal decomposition and feature extraction were carried out for the acquired original vibration signals, using the fault feature extraction method based on MEEMD and AR spectrum. Finally, the fault diagnosis method based on WKELM was used to diagnose the working states of the hydraulic pump. The results show that the diagnosis accuracy rate can reach 100%. Compared with BP, SVM, and ELM, the feasibility and superiority of the method proposed in this paper are demonstrated.

## 2. Basic Principles of Modified Ensemble Empirical Mode Decomposition (MEEMD) and AutoRegressive (AR) Spectrum Method

### 2.1. Basic Principles of MEEMD Method

MEEMD is an improvement on EEMD [29]. First, considering that the set of Gaussian white noise added in EEMD cannot be completely canceled out, MEEMD adds a pair of Gaussian white noise signals instead, where the amplitude of the white noise is the same but the sign is opposite. The purpose of this is to allow the added Gaussian white noise signals to cancel each other out, thereby reducing the influence of noise on the signal [30]. Secondly, considering that EEMD will decompose more pseudo-components, the concept of PE is introduced to MEEMD, in order to detect whether the signal is abnormal. The abnormal signals are decomposed again and eliminated, such that the pseudo-components can be effectively reduced. At the same time, the MEEMD method can also suppress the phenomenon of mode mixing in the EMD and EEMD methods. Therefore, the MEEMD method has a better decomposition effect than EMD or EEMD.

The Specific steps of MEEMD are as follows [14]:

(1) Two groups of Gaussian white noises, *noise_l_*(*i*) and –*noise_l_*(*i*), with equal absolute values, are added to the original signal $\left\{x\left(i\right),i=1,2,\ldots ,n\right\}$:(1)$$\begin{array}{l} x^{+}\left(i\right)=x\left(i\right)+a_{l}noise_{l}\left(i\right)\\ x^{-}\left(i\right)=x\left(i\right)-a_{l}noise_{l}\left(i\right) \end{array}$$
where $noise_{l}\left(i\right)$ is the added Gaussian white noise signal, $a_{l}$ is the amplitude of the added noise signal, and l=1,2,…,Ne, where Ne is the number of white noise additions.

The signals $x^{+}\left(i\right)$ and $x^{-}\left(i\right)$, after adding Gaussian white noise, are decomposed by EMD. The first-order Intrinsic Mode Function (IMF) component sequences are respectively expressed as $\left\{c_{l,1}^{+}\left(i\right)\right\}$ and $\left\{c_{l,1}^{-}\left(i\right)\right\}$, where l=1,2,…,Ne. The overall average of the above components can eliminate the residual white noise to the greatest extent, that is:(2)$$c_{1}\left(i\right)=\frac{1}{2Ne}\sum_{l=1}^{Ne}\left[c_{l1}^{+}\left(i\right)+c_{l1}^{-}\left(i\right)\right].$$

The PE of the IMF components $c_{1}\left(i\right)$ is calculated after the overall average. If the PE is greater than θ, it is considered to be an abnormal signal; otherwise, it is approximately considered to be a stationary signal. The general range of θ is 0.55–0.6; θ=0.6 is taken in this paper.

(2) Determine whether $c_{1}\left(i\right)$ is an abnormal signal. If $c_{1}\left(i\right)$ is an abnormal signal, the first IMF component $c_{1}\left(i\right)$ will be separated from the original signal $x\left(i\right)$, where the new signal $r_{1}\left(i\right)$ is:(3)$$r_{1}\left(i\right)=x\left(i\right)\;-\;c_{1}\left(i\right).$$

Taking the signal $r_{1}\left(i\right)$ as the original signal, Step (1) is repeated until the IMF component $c_{q}\left(i\right)$ obtained after the overall average is not an abnormal signal.

(3) The first q−1 decomposed abnormal signal components whose PE is greater than θ are separated from the original signal, namely:(4)$$r_{q-1}\left(i\right)\;=\;x\left(i\right)\;-\;\sum_{j=1}^{q-1}c_{j}\left(i\right),$$
where j=1,2,…,q−1, q−1 is the number of abnormal components, and $r_{q-1}\left(i\right)$ is the residual signal after removing the abnormal components.

(4) After the remaining signal, $r_{q-1}\left(i\right)$, is decomposed by EMD, the decomposed IMF components are arranged, in order from high frequency to low frequency:(5)$$r_{q-1}\left(i\right)\overset{EMD}{\to }\sum_{k=1}^{z}\left[c_{k}{}^{\prime }\left(i\right)\right]+r^{\prime }\left(i\right).$$

Finally, MEEMD method can be expressed as:(6)$$x\left(i\right)\overset{MEEMD}{\to }\sum_{j=1}^{q-1}c_{j}\left(i\right)+\sum_{k=1}^{z}\left[c_{k}{}^{\prime }\left(i\right)\right]+r^{\prime }\left(i\right),$$
where $c_{k}{}^{\prime }\left(i\right)$ are the IMF components finally obtained, k=1,2,…,z, z is the number of IMF components finally obtained, and $r^{\prime }\left(i\right)$ is the residual component finally obtained.

### 2.2. Permutation Entropy

Permutation entropy (PE) [31] is a method for detecting the sudden change of dynamic behavior and signal regularity, which has the advantages of fast calculation speed and strong anti-interference ability. It is especially suitable for non-linear and non-stationary signal analysis and has good robustness. The steps of the permutation entropy method are as follows:

Given a period of time-series: $\left\{x\left(i\right),i=1,2,\ldots ,n\right\}$, the reconstruction matrix B can be obtained by phase space reconstruction:(7)$$B=\left[\begin{array}{cccc} x(1) & x(1+\lambda ) & \cdots & x(1+(m-1)\lambda )\\ x(2) & x(2+\lambda ) & \cdots & x(2+(m-1)\lambda )\\ \vdots & \vdots & \ddots & \vdots \\ x(k) & x(k+\lambda ) & \cdots & x(k+(m-1)\lambda ) \end{array}\right],$$
where λ is the time delay, m is the embedding dimension, and k=n−(m−1)λ. Each row b(i) in the reconstruction matrix B can be regarded as a reconstruction component.

The elements of the *i*^th^ reconstruction component b(i) in the reconstruction matrix B are arranged in ascending order, according to the size of the value:(8)$$x\left(i+(j_{1}-1)\lambda \right)\leq x\left(i+(j_{2}-1)\lambda \right)\leq \cdots \leq x\left(i+(j_{m}-1)\lambda \right),$$
where $j_{1},j_{2},\ldots ,j_{m}$ are the indices of the column of the reconstructed component element. If $x\left(i+(j_{1}-1)\lambda \right)=x\left(i+(j_{2}-1)\lambda \right)$, they are sorted by the sizes of $j_{1},j_{2}$; that is, when $j_{1}<j_{2}$, $x\left(i+(j_{1}-1)\lambda \right)\leq x\left(i+(j_{2}-1)\lambda \right)$.

Therefore, for any discrete sequence $\left\{x\left(i\right),i=1,2,\ldots ,n\right\}$, a set of symbolic sequences can be obtained for each row of its reconstruction matrix B:(9)$$S(l)=(j_{1},j_{2},\ldots ,j_{m}),$$
where l=1,2,…,k,  k≤m!, as there are *m*! permutations of *m* symbols $\left\{j_{1},j_{2},\ldots ,j_{m}\right\}$ and, so, the *m*-dimensional phase space can map *m*! kinds of symbol sequences, where S(l) is just one of them. The probability $\left\{P_{1},P_{2},\ldots ,P_{k}\right\}$ of each symbol sequence is calculated, following which the PE of the discrete sequence $\left\{x\left(i\right),i=1,2,\ldots ,n\right\}$ can be defined as [32]:(10)$$H_{P}(m)=-\sum_{j=1}^{k}P_{j}lnP_{j}.$$

When $P_{j}=1/m!$, then $H_{P}(m)$ reaches its maximum ln(m!). For convenience, ln(m!) is often used to standardize $H_{P}(m)$. Thus, the PE obtained in Equation (4) can be standardized as follows:(11)$$PE_{norm}=\frac{H_{p}\left(m\right)}{\ln\left(m!\right)},$$
where $PE_{norm}\;\in \;\left[0,1\right]$. The smaller the value of $PE_{norm}$, the more regular the time-series; thus, $PE_{norm}$ can reflect the randomness of the time-series [33].

### 2.3. AR Spectrum Method

The autoregressive (AR) model is a time-series analysis method. It not only can fully reveal the characteristics of a signal, but also can improve the resolution of the signal. By analyzing the signal using an AR model, the characteristic frequency contained in the signal can be effectively extracted, such that feature extraction can be carried out accurately. The AR spectrum method can overcome the limitations of the Hilbert method, as there is neither estimation error nor a windowing effect. Therefore, the AR spectrum has great advantages in fault feature extraction. The general expression of the AR model is [34]:(12)$$x\left(n\right)=u\left(n\right)-\sum_{k=1}^{p}a_{k}x\left(n-k\right),$$
where $a_{k}(k=1,2,\ldots ,p)$ are the parameters of the AR model, p is the order of the model, and $u\left(n\right)$ is a stationary white noise sequence with a mean value of zero and a variance of $\sigma^{2}$.

The AR(*p*) process $\left\{x\left(n\right)\right\}$ defined by Equation (12) can be regarded as a white noise sequence $u\left(n\right)$ generated by an all-pole filter, whose transfer function is:(13)$$H(z)=\frac{1}{1+\sum_{k=1}^{p}a_{k}z^{-k}}.$$

Therefore, the power spectral density, $p_{x}(e^{j\omega })$, of $\left\{x\left(n\right)\right\}$ can be expressed as:(14)$$p_{x}(e^{j\omega })=\sigma^{2}{\left|H(e^{j\omega })\right|}^{2}=\frac{\sigma^{2}}{{\left|1+\sum_{k=1}^{p}a_{k}e^{-jk\omega }\right|}^{2}}.$$

The AR spectrum has the advantages of high resolution, smoothness, ease of locating, and so on. By analyzing a signal through the AR model, the characteristic frequency contained in the signal can be effectively extracted and the feature extraction can be carried out accurately.

## 3. Wavelet Kernel Extreme Learning Machine

WKELM has been proposed as an improvement on the basis of ELM. Considering that a wavelet kernel function has the characteristics of multi-scale approximation by wavelet analysis, it has a better effect for non-linear classification. Therefore, a wavelet kernel function was used to replace the matrix multiplication of the output matrix, H, of the ELM hidden layer neurons. This not only can improve the fitting ability of the method for non-linear samples but can also improve the stability of the method and shorten the training time.

### 3.1. Extreme Learning Machine Theory

The key characteristic of ELM is that the input weights and hidden layer biases of the network are randomly selected [35], and there is no need for iterative adjustment in the process of operating the method. Therefore, the calculation speed of ELM is extremely fast, which can greatly save time and effectively overcome the shortcomings of traditional single-hidden layer feed-forward neural networks, which have slow convergence speed and can easily become stuck in local minima. ELM can minimize the training error in the process of training and ensure that the norm of the output weights is very small, such that it has higher function approximation ability and generalization performance than traditional neural networks.

ELM [18] is a single-hidden layer feed-forward neural network, which is composed of an input layer, a hidden layer, and an output layer. The ELM network structure is similar to that of a single-hidden layer feed-forward neural network, as shown in Figure 1.

Suppose we have an ELM with u neurons in the input layer, v neurons in the hidden layer, and w neurons in the output layer. Given a training sample set $\left(X,T\right)=\left\{\left(x_{j},t_{j}\right)\left|1\leq j\leq \right.M\right\}$ of *M* groups, where the input sample set is $x_{j}={\left[x_{j1},x_{j2},\ldots ,x_{ju}\right]}^{T}$ and the expected output sample set is $t_{j}={\left[t_{j1},t_{j2},\ldots ,t_{jw}\right]}^{T}$, the actual output of the ELM is:(15)$$\sum_{i=1}^{v}\beta_{i}g\left(w_{i}\cdot x_{j}+b_{i}\right)=o_{j}\;\;\;\;\;j=1,2,\cdots ,M,$$
where $w_{i}={\left[w_{i1},w_{i2},\cdots ,w_{iu}\right]}^{T}$ are the weights between the input neurons and the *i*^th^ hidden layer node; $\beta_{i}$ are the weights between the *i*^th^ hidden layer node and the output layer neurons, $\beta_{i}=[\beta_{i1},\beta_{i2},\cdots ,\beta_{iw}{]}^{T}$; $b_{i}$ are the biases of the *i*^th^ hidden layer node; and $g\left(\cdot \right)$ is the activation function of the ELM hidden layer.

When the number of hidden layer neurons, v, and the number of training set samples, *M,* of single-hidden layer feedforward neural network are equal, the neural network can approach the training samples with zero error, that is:(16)$$\sum_{j=1}^{M}{\left\|o_{j}-t_{j}\right\|}_{2}=0.$$

Then, we have $\beta_{i}$, $w_{i}$, and $b_{i}$, such that:(17)$$\sum_{i=1}^{v}\beta_{i}g\left(w_{i}\cdot x_{j}+b_{i}\right)=t_{j}\;\;\;\;j=1,2,\cdots ,M.$$

Define:(18)$$H={\left[\begin{array}{c} h\left(x_{1}\right)\\ \vdots \\ h\left(x_{M}\right) \end{array}\right]}_{M\times v}={\left[\begin{array}{ccc} g(w_{1}\cdot x_{1}+b_{1}) & \cdots & g(w_{v}\cdot x_{1}+b_{v})\\ \vdots & \vdots & \vdots \\ g(w_{1}\cdot x_{M}+b_{1}) & \cdots & g(w_{v}\cdot x_{M}+b_{v}) \end{array}\right]}_{M\times v},$$
(19)$$\beta ={\left[\begin{array}{c} \beta_{1}^{T}\\ \vdots \\ \beta_{v}^{T} \end{array}\right]}_{v\times w},$$
(20)$$T={\left[\begin{array}{c} t_{1}^{T}\\ \vdots \\ t_{v}^{T} \end{array}\right]}_{M\times w}.$$

Therefore, Equation (17) can be written as:(21)Hβ=Τ,
where H is the hidden layer output matrix of ELM.

When the activation function, $g\left(\cdot \right)$, of the hidden layer is infinitely differentiable, the weight matrix W of the input and hidden layers and the bias b of the hidden layer can be randomly determined before training and remain unchanged during training. In this case, the hidden layer output matrix H is a constant matrix. In this way, the solution of the weight matrix β between the hidden layer and the output layer can be transformed into the solution of the least-squares solution $\hat{\beta }$ of the linear equations Hβ=Τ, namely:(22)$$\hat{\beta }=H^{+}T,$$
where $H^{+}$ is the Moore–Penrose generalized inverse of the hidden layer output matrix H.

According to Karush–Kuhn–Tucker (KKT) theory, the solution of Equation (22) can be transformed into the solution of the Lagrangian function [36]:(23)$$\min\;L_{\mathrm{ELM}}=\frac{1}{2}{\left\|\beta \right\|}_{1}^{2}+\frac{C}{2}\sum_{i=1}^{M}{\left\|\xi_{i}\right\|}_{2}^{2}-\sum_{i=1}^{M}\alpha_{i}\cdot \left[h\left(x_{i}\right)\beta -t_{i}+\xi_{i}\right],$$
where β is the output weight vector, C is a penalty factor, $\xi_{i}$ is the training error, $\alpha_{i}$ are Lagrange operators, $h\left(x_{i}\right)$ is the feature mapping of the hidden layer, $t_{i}$ are the target output values, and *M* is the number of samples.

Calculating the partial derivative of Equation (23), the output weights, $\hat{\beta }$, of the ELM can be solved as:(24)$$\hat{\beta }=H^{T}{\left(HH^{T}+\frac{I}{C}\right)}^{-1}T,$$
where I is the identity matrix of order v.

Finally, the output function of the ELM can be expressed as [37]:(25)$$f\left(x\right)=h\left(x\right)\hat{\beta }=h\left(x\right)H^{T}{\left(HH^{T}+\frac{I}{C}\right)}^{-1}T.$$

### 3.2. Theory of Kernel Extreme Learning Machine

As the weights between the input layer and the hidden layer and hidden layer biases of the ELM are randomly selected during training, a good training accuracy cannot be guaranteed every time; even the same training sample can have different results in two training processes. In order to solve this problem, Huang et al. proposed an improved ELM method in 2012; namely, kernel-ELM (KELM). The KELM method introduces a kernel function into the ELM method, which replaces the method of randomly selecting the input weights and hidden layer biases in original ELM method.

In Equation (21), the hidden layer output matrix, H, of ELM can be written as:(26)$$H={\left[\begin{array}{c} h\left(x_{1}\right)\\ \vdots \\ h\left(x_{M}\right) \end{array}\right]}_{M\times v},$$
where $h\left(x_{i}\right)$ is the feature mapping of the hidden layer. $HH^{T}$ in Equation (24) can be replaced by constructing a kernel function without knowing the specific form of $h\left(x_{i}\right)$:(27)$$HH^{T}\left(i,j\right)=K\left(x_{i},x_{j}\right),$$
where $K\left(x_{i},x_{j}\right)$ is the kernel function and i,j∈(1,2,…,M).

The ELM kernel matrix is defined as:(28)$$\Omega_{\mathrm{ELM}}=HH^{T}=\left[\begin{array}{ccc} K\left(x_{1},x_{1}\right) & \cdots & K\left(x_{1},x_{M}\right)\\ \vdots & \vdots & \vdots \\ K\left(x_{M},x_{1}\right) & \cdots & K\left(x_{M},x_{M}\right) \end{array}\right].$$

Then, Equation (25) for the ELM can be changed into the form of KELM:(29)$$f\left(x\right)=\left[\begin{array}{c} K\left(x,x_{1}\right)\\ \vdots \\ K\left(x,x_{M}\right) \end{array}\right]{\left(\Omega_{\mathrm{ELM}}+\frac{I}{C}\right)}^{-1}T.$$

The KELM method was obtained by introducing a kernel method into the ELM method. Compared with basic ELM, the KELM method has better non-linear function approximation ability and superior data classification ability. In addition, KELM does not need to randomly select input weights and hidden layer biases during training, only requiring the selection of appropriate kernel functions to overcome the shortcomings of extreme learning machines. In addition, KELM not only can overcome the dimensional disaster problem of ELM, but also has very high recognition accuracy.

### 3.3. Wavelet Kernel Extreme Learning Machine Theory

The basic use of a wavelet function is to fit any function with wavelets of different scales. As wavelet functions have strong mapping ability, a wavelet kernel can be used as the kernel function in the KELM method [38]. Therefore, a wavelet kernel function is constructed from a wavelet function in this paper, which is used as the kernel function of KELM. Thus, a WKELM is constructed.

Given a mother wavelet function, $\psi \left(x\right)$, whose scale factor and translation factor are a and b respectively, the wavelet basis function can be expressed as:(30)$$\psi_{a,b}\left(x\right)=\frac{1}{\sqrt{\left|a\right|}}\psi \left(\frac{x-b}{a}\right).$$

According to tensor product theory, a multidimensional wavelet function can be expressed by tensor products of several one-dimensional wavelet functions:(31)$$\psi \left(x\right)=\prod_{i=1}^{s}\psi \left(x_{i}\right),$$
where *s* is the number of one-dimensional wavelet functions, when the multidimensional wavelet function is written in tensor product form; and $x_{i}$ is the independent variable of the *i*^th^ one-dimensional wavelet function.

According to Equation (31), a translation-invariant kernel function can be constructed:(32)$$K\left(x,x^{\prime }\right)=\prod_{i=1}^{s}\psi \left(\frac{x_{i}-{x^{\prime }}_{i}}{a}\right).$$

In this paper, the Morlet wavelet is used to construct the wavelet kernel function. The Morlet wavelet function is given as:(33)$$\psi \left(x\right)=\cos\left(1.75x\right)\exp\left(-\frac{x^{2}}{2}\right).$$

The wavelet kernel function constructed from the Morlet wavelet function is:(34)$$K\left(x,x^{\prime }\right)=\prod_{i}^{s}\left[\cos\left(1.75\times \frac{x_{i}-x_{i}^{\prime }}{a}\right)\exp\left(-\frac{{\left(x_{i}-x_{i}^{\prime }\right)}^{2}}{2a^{2}}\right)\right],$$
where $x_{i}$ are the *i*^th^ components of x.

As the wavelet kernel function is constructed using the translation-invariant kernel of the wavelet function, the wavelet kernel function inherits the multi-scale approximation characteristics of the wavelet function. Therefore, taking a wavelet kernel function as the kernel function in KELM will have a better effect on classification. The method flow is as follows:(1)The number of input neurons and output neurons are determined, according to the number of features and categories, and the WKELM classifier model is established;(2)Set the value of the penalty factor *C* and the parameter of the wavelet kernel *a*;(3)The input characteristics of the training samples and their corresponding output categories are used to train the WKELM;(4)The features of the test sample are used as the input for the WKELM and, then, the trained WKELM classifier is used for the category decision.

## 4. Fault Feature Extraction Method of Axial Piston Pump Based on MEEMD and AR Spectrum

### 4.1. Test System and Data Acquisition

A hydraulic pump fault simulation test bed was used for the test. The vibration sensor used in this paper is piezoelectric and charge output type vibration sensor. The charge amplifier is used to amplify and convert the output charge of the vibration sensor, which is used to sample the vibration signal of the hydraulic pump shell in *x*, *y* and *z* directions. The pressure sensor is a piezoresistive pressure sensor, which is used to sample the pressure pulsation signal of the hydraulic oil at the outlet of the hydraulic pump. The main components used in the test are described in Table 1.

The installation of an experimental installation is shown in Figure 2. The vibration sensors are used to collect the vibration signals of the hydraulic pump in different states. The schematic diagram of the test system is shown in Figure 3. During the test, the different fault states of the hydraulic pump were simulated by replacing the plunger and slipper components of different fault types, as well as through use of a worn center spring. The outlet pressure of the hydraulic pump could be controlled by adjusting the pilot-operated proportional overflow valve 18. Vane pump 3 was used as a supplementary pump, in order to make up for the poor self-priming capacity of the piston pump.

During the test, a data acquisition system was required to display and store the vibration signal collected by the vibration sensor in real-time [39]. The data acquisition system used in this test was based on the LabVIEW software (National Instruments (NI), Austin, Texas, USA). As shown in Figure 4, the sampling frequency, sampling time, and controls for starting and stopping the data collection could be set using the front panel. At the same time, the collected pump speed, pressure, vibration, and other information could be displayed in real-time.

The fault setting methods of the axial piston pump during the test are shown in Table 2. The process of the test was as follows: First, the three vibration sensors were fixed to the end cap and shell of the hydraulic pump, respectively, as shown in Figure 3. When the hydraulic system was unloaded, the test equipment was started. When the system reached normal speed, the system pressure was set to 10 MPa through the pilot-operated proportional overflow valve 18. The speed of the motor was set to 1480 r/min. Then, after the sampling frequency was set to 20 kHz in the data acquisition system, the data acquisition system was started, in order to collect and store the speed and other signals of the hydraulic pump in the normal state. Finally, the normal components were replaced by the fault components of the axial piston pump, and the vibration signal of the hydraulic pump in the fault state was acquisited. After the data acquisition stage, the original signals of the hydraulic pump in normal state and three fault states were obtained. Some pictures of the fault components are shown in Figure 5.

### 4.2. Simulation sSignal Analysis of MEEMD Method

A simulation signal, $x\left(t\right)$, with a sampling frequency of 1000 Hz and a sampling time of 2s was constructed:(35)$$x\left(t\right)=x_{1}\left(t\right)+x_{2}\left(t\right)+n\left(t\right),$$
where $x_{1}\left(t\right)$ is a sine signal, $x_{2}\left(t\right)$ is an amplitude modulation signal, and $n\left(t\right)$ are two intermittent random signals:(36)$$x_{1}\left(t\right)=2\sin\left(2\pi \cdot 30t+\frac{\pi }{2}\right),$$
(37)$$x_{2}\left(t\right)=\left(t+1\right)2\sin\left(2\pi \cdot 8t+\frac{\pi }{3}\right).$$

Figure 6 shows the time-domain diagram of the synthesized signal and its components. The synthesized signal was decomposed by EEMD, and the decomposition results obtained are shown in Figure 7.

The amplitude of the added Gaussian white noise was $a_{l}=0.2$, and the total number of noises added was *N_e_* = 100. The results obtained by EEMD decomposition of the signal are shown in Figure 7. It can be seen, from Figure 7, that EEMD decomposed many pseudo-components. After EEMD decomposition of the simulation signal *x*(*t*), seven IMF components and one residual component were decomposed, among which IMF3 and IMF5, respectively, represent the sine signal *x*_1_(*t*) and the amplitude modulation signal *x*_2_(*t*) in the synthetic simulation signal *x*(*t*). The simulation signal *x*(*t*) was also decomposed by MEEMD, as shown in Figure 8. It can be seen, from Figure 3, that three IMF components and one residual component were decomposed, among which IMF2 and IMF3, respectively, represent *x*_1_(*t*) and *x*_2_(*t*) in the synthetic simulation signal *x*(*t*). Compared with EEMD, MEEMD can effectively eliminate the pseudo-components.

### 4.3. Fault Feature Extraction Process

In this paper, a fault feature extraction method based on MEEMD and AR spectrum is proposed. First, the non-stationary vibration signals are decomposed by MEEMD, in order to obtain some IMF components. Secondly, the effective IMF components are screened by the Pearson correlation coefficient method, then analyzed by the AR spectrum method. Finally, the calculated AR spectrum energy is taken as the fault feature of the signal. Compared with the Hilbert-Huang transform (HHT) method, the feature extraction method based on MEEMD and AR spectrum proposed in this paper not only can solve the mode mixing phenomenon of EMD decomposition by the HHT method but can also overcome the limitation of Hilbert analysis in HHT method. The specific steps of this method are as follows:(1)The signal is decomposed by the MEEMD method, from which several IMF components are obtained.(2)As there may still be pseudo-components in the IMF components obtained through MEEMD decomposition, it is necessary to eliminate such components. In this paper, the pseudo-components are eliminated by the Pearson correlation coefficient method. First, the Pearson correlation coefficient r between all IMF components and the original signal is calculated. Then, t IMF components with large correlation coefficients are taken as the effective components. The Pearson correlation coefficient r is calculated as follows:(38)$$r=\frac{\sum_{i=1}^{n}\left[x\left(i\right)-\bar{x}\right]\left[y\left(i\right)-\bar{y}\right]}{\sqrt{\sum_{i=1}^{n}{\left[x\left(i\right)-\bar{x}\right]}^{2}}\sqrt{\sum_{i=1}^{n}{\left[y\left(i\right)-\bar{y}\right]}^{2}}},$$
where $x\left(i\right)$ is the original signal, $y\left(i\right)$ is each order of IMF components, i=1,2,⋯,n, n is the signal length, $\bar{x}$ is the average of $x\left(i\right)$, and $\bar{y}$ is the average of $y\left(i\right)$.(3)AR spectrum estimation of the *t* IMF components is carried out and the AR spectrum energies of the *t* IMF components are calculated. After normalization, the characteristic matrix, P, of the signal is composed as follows:(39)$$P={\left[p_{1},p_{2},\cdots ,p_{M}\right]}_{t\times M},$$
where $p_{i}={\left[p_{1i},p_{2i},\cdots ,p_{ti}\right]}^{T}$, i=1,2,⋯,M, and $p_{ki}$ is the AR spectrum energy of the *t*^th^ IMF component of the *i*^th^ sample.


### 4.4. Analysis of Acquired Hydraulic Pump Signals 

The *x*-direction vibration signal of a single slipper loosening fault was selected for analysis. Figure 9 shows the time-domain diagram of the vibration signal in the state of the single slipper loosening fault of the axial piston pump.

In order to show that the MEEMD method had a better effect than the EEMD method in the signal processing of the hydraulic pump, the number of IMF components and the index of orthogonality (*IO*) decomposed by the two methods were compared. Figure 10 shows the vibration signal decompositions of hydraulic pump in the single slipper loosing fault state by the two methods. Table 3 gives the *IO* of EEMD and MEEMD methods. Orthogonality refers to the correlation between different components. The orthogonality of the signal can be measured by calculating the *IO*, after which the degree of modal mixing between the components can be judged. The larger the *IO* value, the worse the orthogonality; that is, the more serious the degree of modal mixing. The Equation for *IO* of IMF components is [11]:(40)$$IO=\sum_{i=0}^{n}\left[\sum_{j=1}^{I}\sum_{k=1}^{I}\frac{c_{j}\left(i\right)c_{k}\left(i\right)}{x^{2}\left(i\right)}\right],$$
where j≠k, $c_{j}\left(i\right)$ is the *j*^th^ IMF component, $c_{k}\left(i\right)$ is the *k*^th^ IMF component, $x\left(i\right)$ is the original sequence, n is the length of the time-series, I is the number of IMF components, j=1,2,⋯,I, and k=1,2,…,I.

It can be seen, from Figure 10, that nine IMF components were decomposed by the EEMD method, while six IMF components were decomposed by the MEEMD method. It is obvious that the MEEMD method addressed the problem of excessive false modal components, as was obvious in the EEMD method. It can be seen, from Table 3, that the IO value for the MEEMD method was smaller than that of the EEMD method. This indicates that the MEEMD method is more effective than the EEMD method in suppressing the phenomenon of modal mixing.

### 4.5. Fault Feature Extraction of Hydraulic Pump Based on MEEMD-AR Spectrum

First, the *x*-direction vibration signals of the hydraulic pump in four states were decomposed by the MEEMD method, and several IMF components were obtained. The vibration signals of the hydraulic pump in normal working state and single slipper wear fault state were decomposed by MEEMD, respectively. The decomposition results are shown in Figure 11.

The Pearson correlation coefficient method was used to calculate the correlation coefficient between all IMF components decomposed by MEEMD and the original signal. Figure 12 shows the mean value of correlation coefficients between each IMF component decomposed by the MEEMD method and the original signal of 50 samples in each of the four hydraulic pump states. It can be seen, from Figure 12, that the correlation coefficient between the first IMF component and the original signal was more than 0.75, the correlation coefficient between the second IMF component and the original signal was basically more than 0.4, the correlation coefficient between the third IMF component and the original signal was more than 0.34, and the correlation coefficient between the fourth IMF component and the original signal is more than 0.3. The correlation coefficients between the other IMF components and the original signal were all less than 0.3.

Considering the complexity and accuracy of the method, the components whose correlation coefficient between the IMF component and original signal was greater than 0.3 were selected as the effective components in this paper. It can be seen, from the analysis of Figure 12, that the correlation coefficients between the first four IMF components and the original signal in the four states of the hydraulic pump were greater than 0.3. Therefore, AR spectrum analysis was only performed for the first four IMF components decomposed by MEEMD in various states, as shown in Figure 13.

It can be seen from Figure 13 that the AR spectra of the same-order IMF components of hydraulic pumps in the four states were different, where there were different characteristics among the different IMF components. In addition, the maps obtained by AR spectrum analysis were not only smooth, but also easy to distinguish. The AR spectrum energies of different states (i.e., corresponding to each IMF component in Figure 12) were calculated, and the results are shown in Table 4.

It can be intuitively seen, from Table 4, that the feature values extracted by different states had different distributions, demonstrating the effectiveness of the feature extraction method, to a certain extent.

## 5. Fault Diagnosis Method of Hydraulic Pump Based on WKELM

Equipment will inevitably fail in the process of operation. In order to identify the equipment state information from the vibration signal, a fault diagnosis method based on the integration of MEEMD, AR spectrum, and WKELM methods is proposed in this paper, in order to identify the equipment state. Figure 14 shows the fault diagnosis flow chart, based on the MEEMD–AR–WKELM integration method.

### 5.1. Hydraulic Pump Fault Diagnosis Based on WKELM

The *z*-direction vibration signal of the axial piston pump was selected for analysis. A sample was intercepted every 2000 data points of the hydraulic pump vibration signal in four states, where 50 samples were taken in each state. The training sample set was composed of 30 samples selected from 50 samples in each state, while the test sample set was composed of the remaining 20 samples. Two groups of AR spectrum energies in four different states of the hydraulic pump were taken to draw the AR spectrum energy distribution diagram of IMF components in different states, as shown in Figure 15.

It can be seen from Figure 15 that the features extracted from the different hydraulic pump working states had different distributions, while the features taken from the same working state had a similar distribution, thus indicating that the feature extraction method had a good effect.

The features of 200 samples in the four states of the axial piston pump were extracted, and the 4-dimensional fault features of each sample were extracted. A total of 30 samples were selected from 50 samples in each state, in order to form the training sample matrix, $P_{120\times 4}^{train}$, while the remaining 20 samples were used to form the test sample matrix, $P_{80\times 4}^{test}$. After the extraction of feature vectors, the WKELM method was used to identify the state of the hydraulic pump. The number of input neurons was u=4 and the number of output neurons was w=4 for the WKELM method. In consideration of the influence of the WKELM parameters on diagnosis accuracy, the penalty factor C and wavelet kernel coefficient a were analyzed. The influences of *C* and *a* on the recognition accuracy of WKELM are shown in Figure 16.

It can be seen from Figure 16, when *C*=10, the training accuracy reaches 100%. When a≤1, and it gradually decreases and then reaches a plateau. When a>1, the test accuracy first increases and then decreases, reaching the maximum 98% at a=1. When *C* = 100, the training accuracy reaches 100%. When a≤2, and it gradually decreases. When a>2, the test accuracy first increases and then decreases, reaching the maximum 99% at a=1.5. When *C* = 1000, the training accuracy is 100%; when a≤4.5, the test accuracy first increases and then decreases, reaching the maximum 98% at a=1.5. When *C* = 10000, although the training accuracy is generally high, the test accuracy is very low. In general, when the value range of a is between 1 and 2.5, the training accuracy and the test accuracy are relatively high. After taking values within this range many times, we finally found that when a=1.9 and *C* = 500, both the training accuracy and the test accuracy reached 100%. Therefore, the parameters of WKELM are determined as a=1.9 and *C* = 500.

After determining the parameters of WKELM, the WKELM model was trained. The characteristic matrix, $P_{120\times 4}^{train}$, of the training samples was taken as the input of WKELM, while the labels corresponding to each state were taken as the output (the label for normal working was 1, the label for single slipper wear fault was 2, the label for single slipper loosing fault was 3, and the label for center spring worn fault was 4). Finally, the characteristic matrix, $P_{80\times 4}^{test}$, of the test samples was tested. The obtained recognition results are shown in Figure 17.

It can be seen, from Figure 17, that all the predicted values identified by WKELM were the same as the true values; that is, the accuracy of the recognition results was 100%. This demonstrates that the proposed fault diagnosis method, based on the integration of MEEMD, AR spectrum, and WKELM methods, is effective.

### 5.2. Comparison and Analysis of Different Hydraulic Pump Fault Diagnosis Methods

In order to prove that WKELM has more advantages than the BP neural network, SVM, and ELM methods, hydraulic pump fault diagnosis was carried out using these methods under the same conditions. Among them, the number of neurons in the input, hidden, and output layers of BP neural network were four, 15, and four, respectively, and the activation function was the Sigmoid function. The LibSVM toolkit, provided by the Taiwanese scholar Lin Chih-Jen, was selected for SVM. The number of hidden layer neurons for the ELM was 14 and the activation function was the sigmoid function. The diagnosis accuracies of the four fault diagnosis methods are shown in Table 5. The training and testing times of the four methods were also compared, as shown in Table 6.

It can be seen from Table 5 and Table 6 that in terms of fault recognition accuracy, WKELM had the highest fault recognition accuracy, followed by the BP neural network and ELM, while the SVM had the lowest fault recognition accuracy. In terms of time, WKELM had the shortest time, followed by ELM, while the BP neural network took the longest time. It can be seen that WKELM performed better than the other three methods, in terms of fault recognition accuracy and time consumption. 

The Case Western Reserve University dataset was used to demonstrate the fault diagnosis method proposed in this paper has higher fault recognition accuracy. The sampling frequency of the data is 12 kHz, the model of the bearing is SKF6205, the damage diameter is 0.36 mm, the motor loads is 2 horsepower. Take the normal working state, rolling ball fault state, inner race fault state, and outer race fault data, 0.2 s for each state data as a sample, take 50 samples under each state, of which 30 as the training sample, 20 as test samples, the results as shown in Table 7. It can be seen that the fault diagnosis method proposed in this paper has a high recognition accuracy.

In order to prove that the advantages of the fault diagnosis method proposed in this paper, hydraulic pump fault diagnosis was carried out using these methods under the same conditions. The diagnosis accuracies of the fault diagnosis methods [40,41] are shown in Table 8. Therefore, the advantages of the fault diagnosis method proposed in this paper, based on the integration of MEEMD, AR spectrum, and WKELM, were proved.

## 6. Conclusions

In this paper, an axial piston pump fault diagnosis method based on MEEMD, AR spectrum, and WKELM integration was proposed and studied. Our main conclusions are as follows:(1)Compared with the EEMD method, the MEEMD method can better suppress the phenomenon of mode mixing in hydraulic pump vibration signal decomposition. At the same time, the MEEMD method has better orthogonality and fewer pseudo-components;(2)The hydraulic pump fault feature extraction method based on the MEEMD and AR spectrum methods proposed in this paper can effectively solve the problem of low efficiency inherent to traditional feature extraction methods. This method can effectively extract the fault features from the vibration signals of a hydraulic pump;(3)The WKELM method was improved and introduced for the fault diagnosis of a hydraulic pump, and a full hydraulic pump fault diagnosis method was proposed based on MEEMD–AR–WKELM integration. The diagnosis accuracy of different hydraulic pump fault states using this method was 100%. Compared with the BP, SVM, and ELM methods, the fault diagnosis method proposed in this paper was shown to have higher fault identification accuracy and faster identification speed.

## Figures and Tables

**Figure 1 sensors-21-02599-f001:**
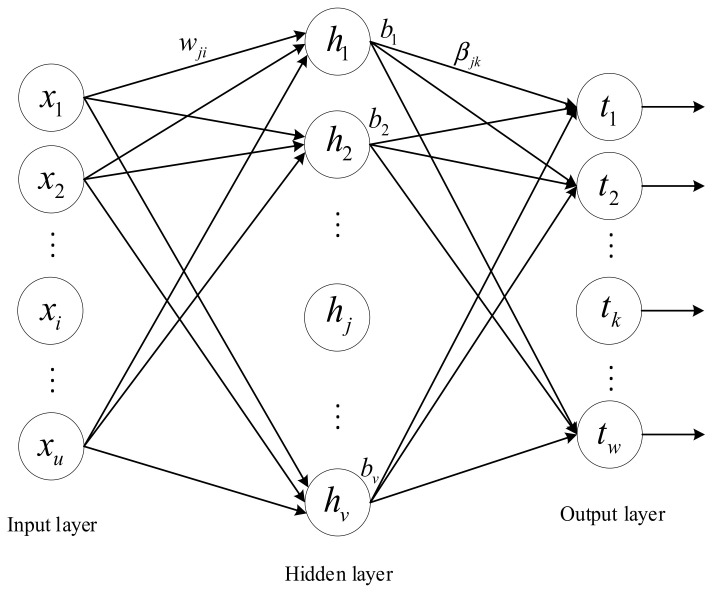
Structure of a single-hidden layer feed-forward neural network.

**Figure 2 sensors-21-02599-f002:**
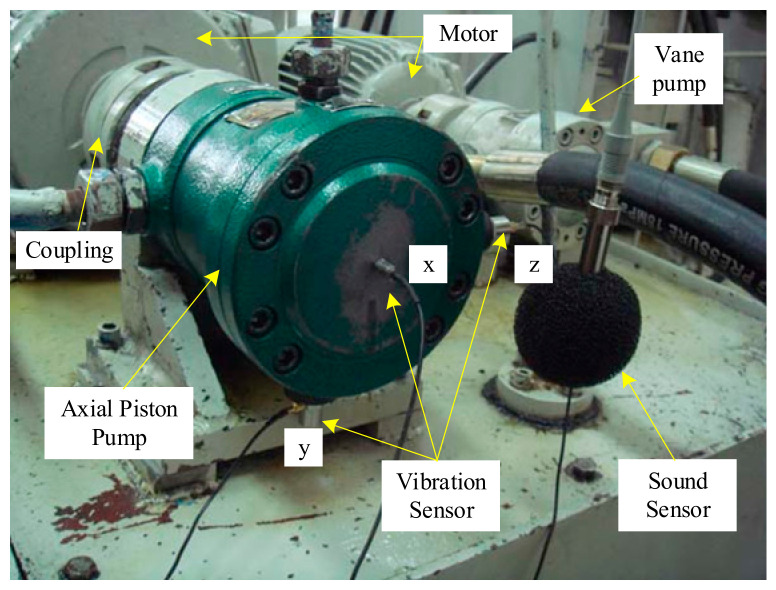
Installation of an experimental installation.

**Figure 3 sensors-21-02599-f003:**
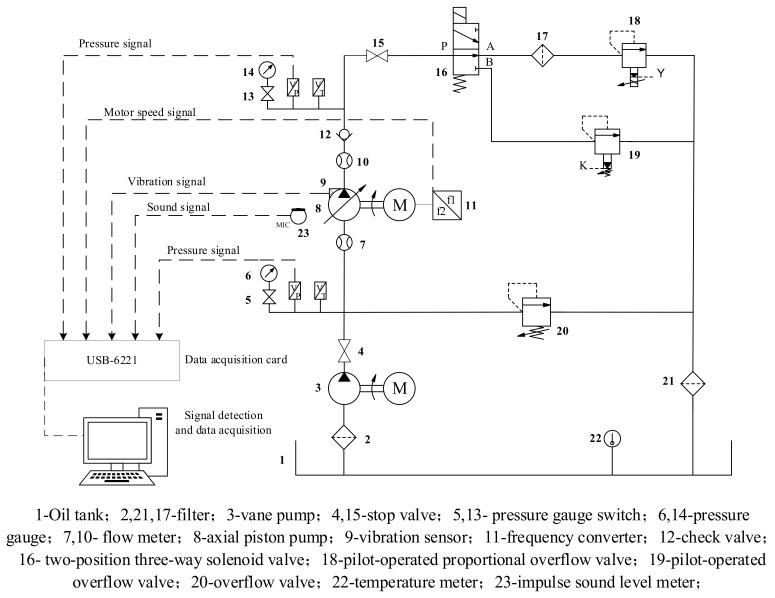
Schematic diagram of the test system.

**Figure 4 sensors-21-02599-f004:**
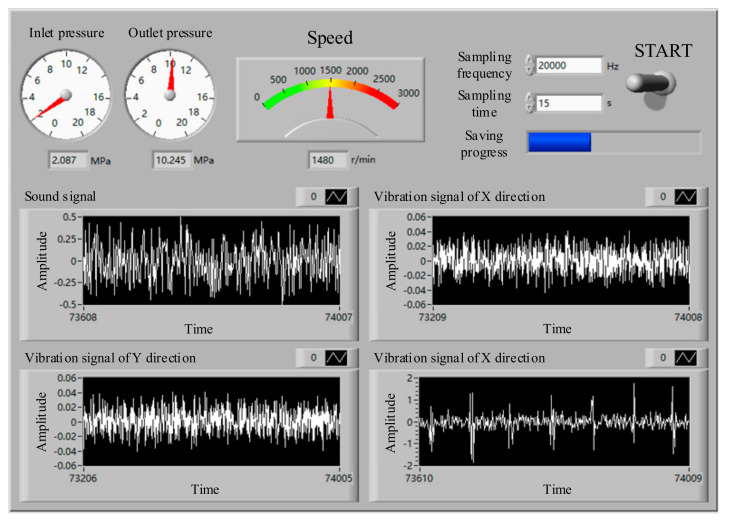
Front panel of the LabVIEW program.

**Figure 5 sensors-21-02599-f005:**
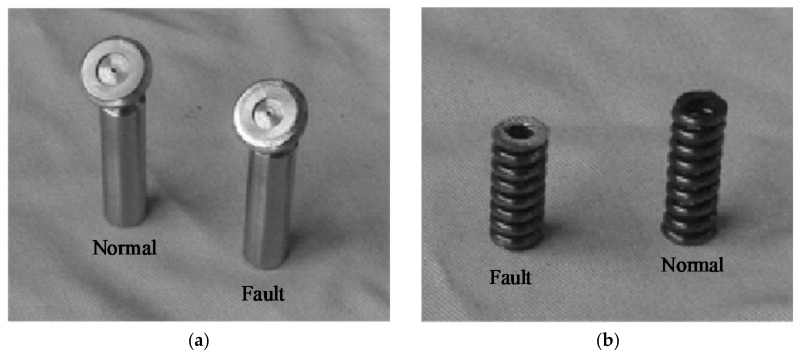
Some pictures of the fault elements: (**a**) Single slipper wear fault; and (**b**) Center spring worn fault.

**Figure 6 sensors-21-02599-f006:**
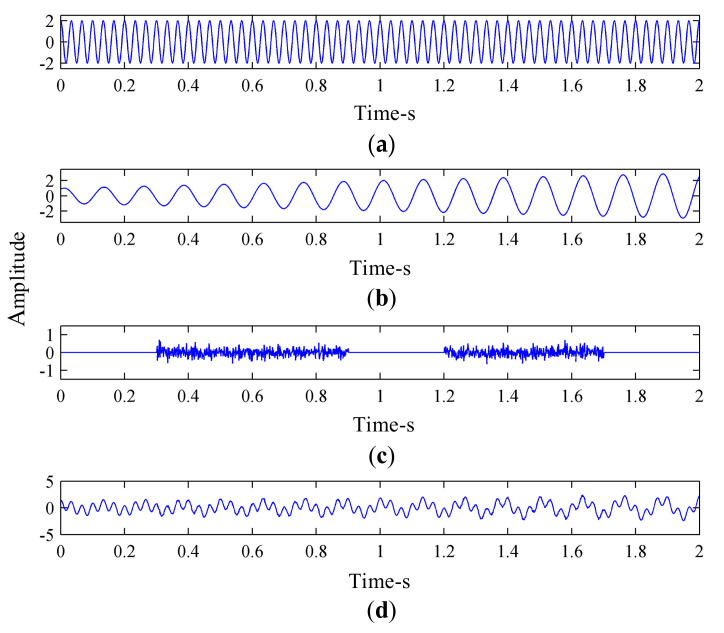
Time-domain diagram of the synthesized signal and its components: (**a**) Sine signal *x*_1_(*t*); (**b**) Amplitude modulation signal *x*_2_(*t*); (**c**) Intermittent random signals *n*(*t*); and (**d**) Synthetic simulation signal *x*(*t*).

**Figure 7 sensors-21-02599-f007:**
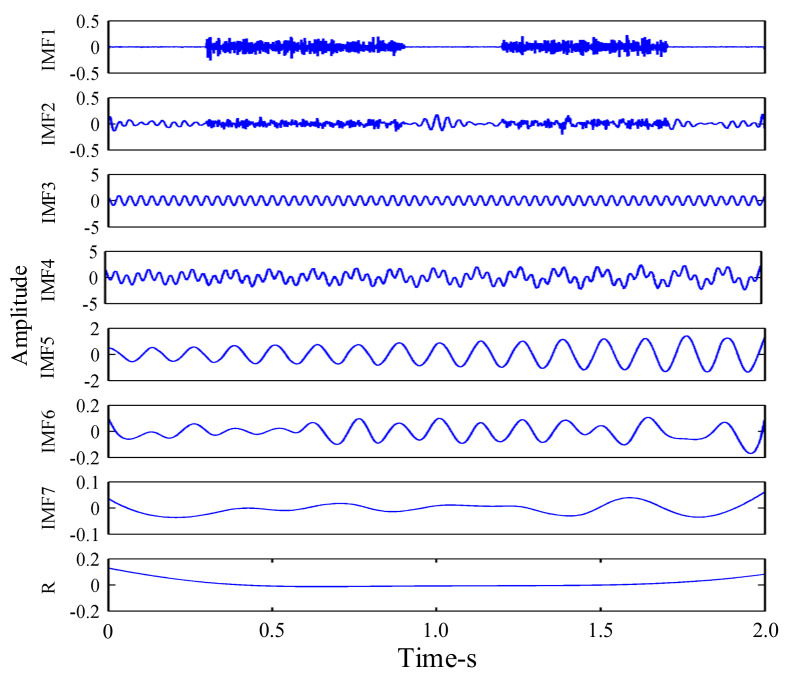
Ensemble empirical mode decomposition (EEMD) decomposition of the synthesized signal.

**Figure 8 sensors-21-02599-f008:**
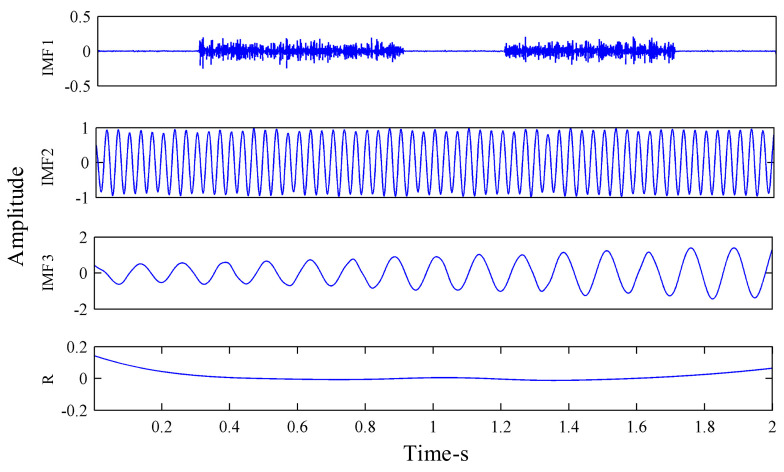
Modified ensemble empirical mode decomposition (MEEMD) decomposition of the synthesized signal.

**Figure 9 sensors-21-02599-f009:**
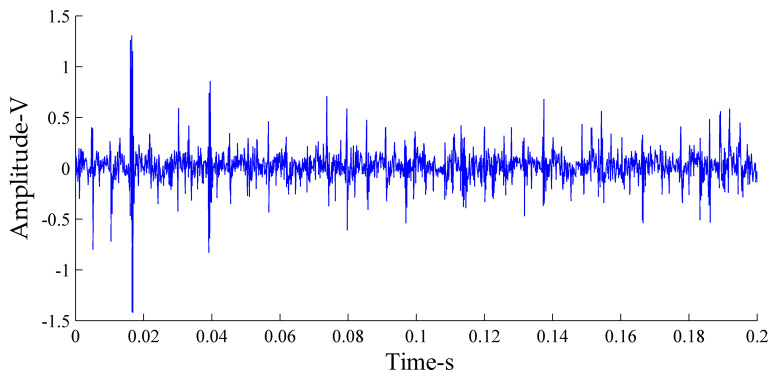
Time-domain diagram of the vibration signal in the single slipper loosing fault state.

**Figure 10 sensors-21-02599-f010:**
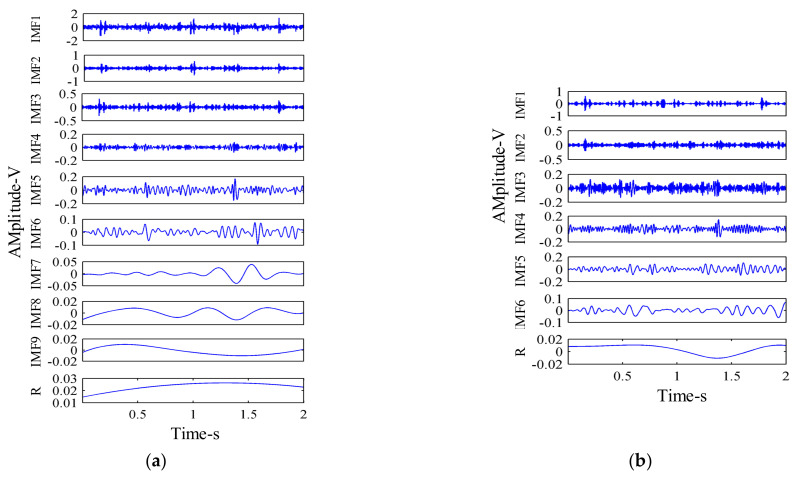
Vibration signal decomposition of hydraulic pump: (**a**) EEMD; and (**b**) MEEMD.

**Figure 11 sensors-21-02599-f011:**
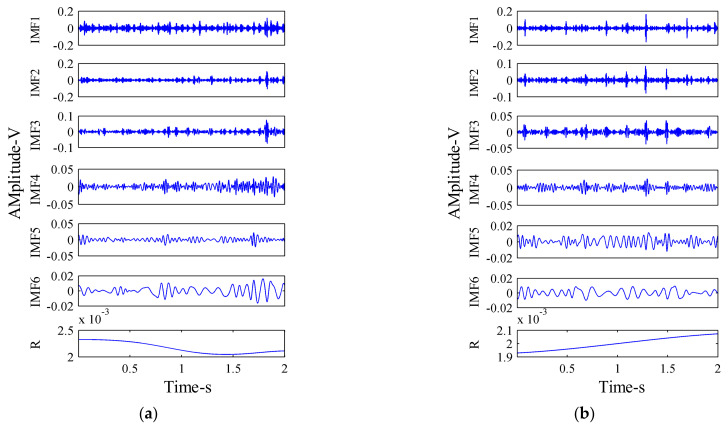
MEEMD decomposition results of vibration signal: (**a**) Normal working state; and (**b**) Single slipper wear fault state.

**Figure 12 sensors-21-02599-f012:**
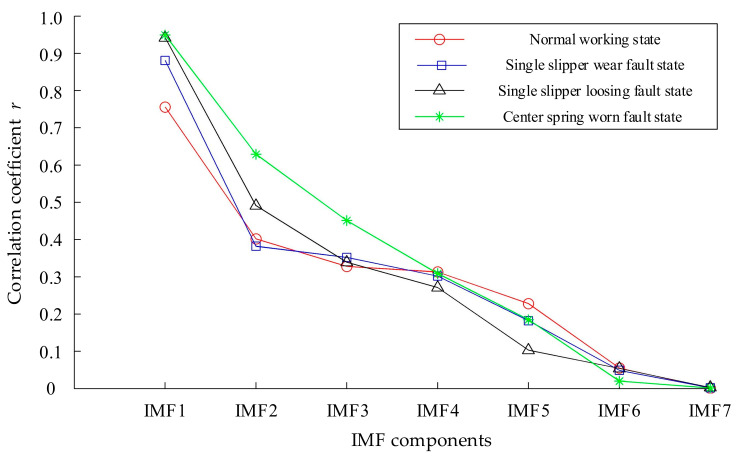
Correlation coefficients between intrinsic mode function (IMF) components of hydraulic pump and original signal.

**Figure 13 sensors-21-02599-f013:**
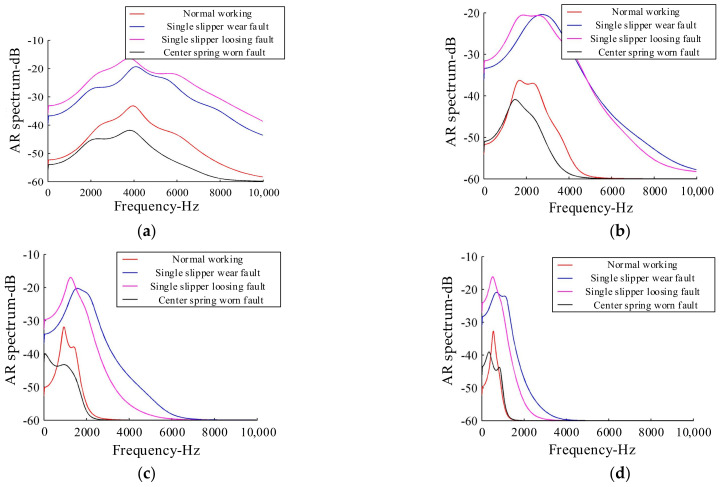
Autoregressive (AR) power spectra of the first four order IMF components: (**a**) IMF 1; (**b**) IMF 2; (**c**) IMF 3; and (**d**) IMF 4.

**Figure 14 sensors-21-02599-f014:**
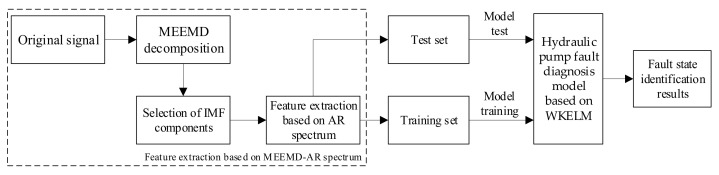
Fault diagnosis flow chart based on the MEEMD–AR–WKELM integration method.

**Figure 15 sensors-21-02599-f015:**
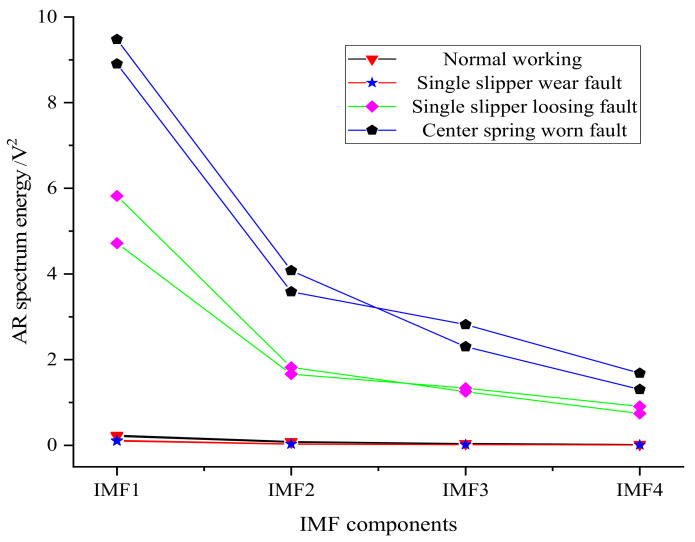
AR spectrum energy distribution of IMF components in different states.

**Figure 16 sensors-21-02599-f016:**
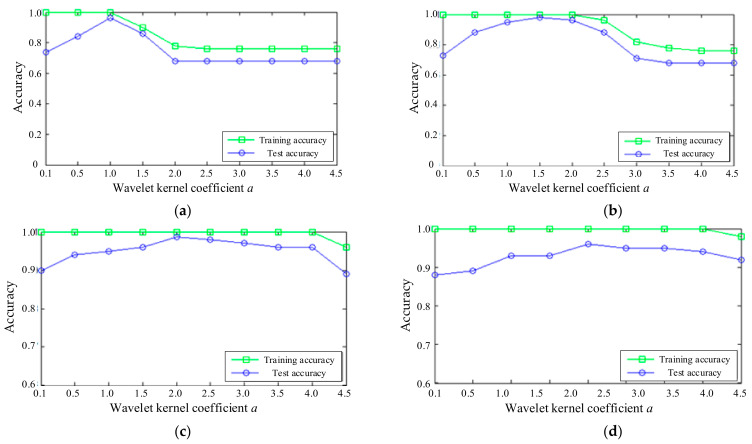
The influence of parameters on accuracy: (**a**) *C* = 10; (**b**) *C* = 100; (**c**) *C* = 1000; and (**d**) *C* = 10,000.

**Figure 17 sensors-21-02599-f017:**
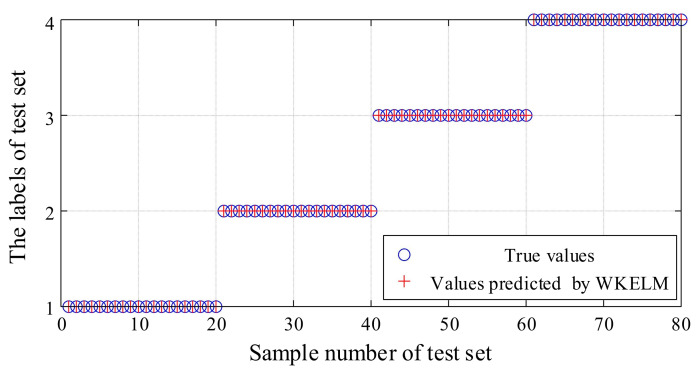
Fault diagnosis of hydraulic pump based on wavelet kernel extreme learning machine (WKELM).

**Table 1 sensors-21-02599-t001:** Types and performance parameters of test components.

Num.	Name	Model	Performance Parameters
1	Motor	Y132M-4	Rated speed: 1480 rpm
2	Axial piston pump	MCY14-1B	Theoretical displacement: 10 mL/r rated pressure: 31.5 MPa; 7 pistons rated speed: 1500 rpm
3	Data acquisition card	USB-6221	Maximum sampling rate: 250 kS/s
4	Vibration sensor	YD-72D	Frequency range: 0.3 Hz–18 kHz Charge sensitivity: 0.35 pC/(m/s^2^) Linear range of amplitude: 1000 m/s^2^
5	Charge amplifier	DHF-10	Maximum output voltage: ±10 V Gain: 0.1m V/pC–1 V/pC; Precision: < 1.5%
6	Pressure Sensor	SYB-351	Measuring range: 0–25 MPa Precision: 0.2%; Output range: 0–5 V
7	Rotational speed measurer	LT-XSMP	Measuring range: 6–45000 rpm

**Table 2 sensors-21-02599-t002:** Fault setting method of axial piston pump.

Num.	States	Fault Setting Method
1	Normal working	---
2	Single slipper wear fault	Grind off a rounded corner of the slipper
3	Single slipper loosing fault	Replace the normal components with a slipper loosing fault components
4	Center spring worn fault	Grind off the center spring by 1.2 mm

**Table 3 sensors-21-02599-t003:** Comparison of Ensemble empirical mode decomposition (EEMD) and Modified ensemble empirical mode decomposition (MEEMD) indicators.

Method	*IO*	Number of IMF Components
EEMD	0.2069	9
MEEMD	0.1131	6

**Table 4 sensors-21-02599-t004:** Autoregressive (AR) spectrum energies of intrinsic mode function (IMF) components in different states.

State	IMF1	IMF2	IMF3	IMF4
Normal working	0.2102	0.0712	0.0374	0.0210
Single slipper wear fault	0.1156	0.0316	0.0151	0.0106
Single slipper loosing fault	4.7201	1.6651	1.3350	0.9085
Center spring worn fault	8.9050	3.5852	2.8181	1.6852

**Table 5 sensors-21-02599-t005:** Comparison of diagnostic accuracy of four fault diagnosis methods.

Condition of Hydraulic Pump		Normal Working	Single Slipper Wear Fault	Single Slipper Loosing Fault	Center Spring Worn Fault	Total
MEEMD–BP	Number of samples correctly diagnosed	20	20	20	19	79
	Recognition accuracy	100%	100%	100%	95%	98.75%
MEEMD–SVM	Number of samples correctly diagnosed	20	19	19	20	78
	Recognition accuracy	100%	95%	95%	100%	97.5%
MEEMD–ELM	Number of samples correctly diagnosed	20	19	20	20	79
	Recognition accuracy	100%	95%	100%	100%	98.75%
MEEMD–WKELM	Number of samples correctly diagnosed	20	20	20	20	80
	Recognition accuracy	100%	100%	100%	100%	100%

**Table 6 sensors-21-02599-t006:** Comparison of training time and testing time of the four fault diagnosis methods.

Fault Diagnosis Method	MEEMD–WKELM	MEEMD–ELM	MEEMD–SVM	MEEMD–BP
Training time (s)	0.0020	0.0054	0.058	0.749
Testing time (s)	0.0011	0.0019	0.0067	0.0042

**Table 7 sensors-21-02599-t007:** Fault diagnosis of rolling bearings based on wavelet kernel extreme learning machine (WKELM).

Condition of Rolling Bearing	Normal Working	Inner Race Fault	Rolling Ball Fault	Outer Race Fault	Total
Number of samples correctly diagnosed	20	20	20	20	80
Recognition accuracy	100%	100%	100%	100%	100%

**Table 8 sensors-21-02599-t008:** Comparison of diagnostic accuracy of fault diagnosis methods.

Fault Diagnosis Method	MEEMD–WKELM	IMSE-KFCM [40]	RQA-KFCM [41]
**Recognition accuracy**	100%	98.75%	98%

## Data Availability

The data presented in this study are available on request from the corresponding author.

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
