# Peer review of "A Hydraulic Pump Fault Diagnosis Method Based on the Modified Ensemble Empirical Mode Decomposition and Wavelet Kernel Extreme Learning Machine Methods"

_sensors, 2021, doi:10.3390/s21082599_

Round 1

Reviewer 1 Report

Comments and specific recommendations:

In this article, a fault diagnosis method for a hydraulic axial piston pump based on the integration of the MEEMD, AR spectrum energy, and WKELM methods is proposed and studied. First, the vibration signals of an axial piston pump under different working states were acquisited from a hydraulic pump fault simulation test bed. Secondly, signal decomposition and feature extraction were carried out for the acquisited original vibration signals, using the fault feature extraction method based on MEEMD and AR spectrum. Finally, the fault diagnosis method based on WKELM was used to diagnose the working states of the hydraulic pump. The results show that the diagnosis accuracy rate can reach 100%. Compared with BP, SVM, and ELM, the feasibility and superiority of the method proposed in this paper are demonstrated. It is suitable for publication after modification. Listed below are some comments that might be considered beforehand:

  • The results were not at all compared with the research of other authors. It is difficult to evaluate the correctness of the experiments and the results without comparison. It is necessary to find as close as possible research oriented in terms of materials and parameters, because in the present form it is only the presentation of the results,

  • the technical principles, resolutions and uncertainty of the test components are not stated (vibration and pressure sensor). Please correct.

  • Figure 5. Physical diagram of the test system - is illegible, and rather Figure 5. Installation of an experimental installation,
  • All cited sources concern only to Introduction part, none are assigned to the discussion or comparison of the results. Add more references with similar issues and focus of journal Sensor (which are totally lacking),
  • The results were not presented on the abstract. The authors needs to describe the main results in the abstract. Please correct.

Author Response

Response to Reviewer 1 Comments

You can  also see the attachment.

Thank you for the reviewers' comments concerning our manuscript entitled "Hydraulic pump fault diagnosis method based on MEEMD and WKELM" (sensors-1160355). Those comments are all valuable and very helpful for revising and improving our paper, as well as the important guiding significance to our researches. We have studied comments carefully and have made correction which we hope meet with approval. Revised portion are marked in yellow in the paper. The main corrections in the paper and the responds to the reviewer's comments are as flowing:

Point 1: The results were not at all compared with the research of other authors. It is difficult to evaluate the correctness of the experiments and the results without comparison. It is necessary to find as close as possible research oriented in terms of materials and parameters, because in the present form it is only the presentation of the results.

Response 1: To solve the problem that the results were not at all compared with the research of other authors, the following contents are added in this paper:

In order to prove that the advantages of the fault diagnosis method proposed in this paper, hydraulic pump fault diagnosis was carried out using these methods under the same conditions. The diagnosis accuracies of the fault diagnosis methods [40-41] are shown in Table 7. Therefore, the advantages of the fault diagnosis method proposed in this paper, based on the integration of MEEMD, AR spectrum, and WKELM, were proved.

Table 7. Comparison of diagnostic accuracy of fault diagnosis methods.

Fault diagnosis method

MEEMD–WKELM

IMSE-KFCM[40]

RQA-KFCM[41]

Recognition accuracy

100%

98.75%

98%

Point 2: The technical principles, resolutions and uncertainty of the test components are not stated (vibration and pressure sensor). Please correct.

Response 2: Aiming at the technical principles, resolutions and uncertainty of the vibration and pressure sensor, the following contents are added in this paper:

The hydraulic pump fault simulation test bed was used for the test. The vibration sensor used in this paper is piezoelectric and charge output type vibration sensor. The charge amplifier is used to amplify and convert the output charge of the vibration sensor, which is used to sample the vibration signal of the hydraulic pump shell in x, y and z directions. The pressure sensor is a piezoresistive pressure sensor, which is used to sample the pressure pulsation signal of the hydraulic oil at the outlet of the hydraulic pump. The main components used in the test are described in Table 1:

Table 1: Types and performance parameters of test components.

Num

Name

Model

Performance parameters

1

Motor

Y132M-4

Rated speed: 1480 rpm

2

Axial piston pump

MCY14-1B

Theoretical displacement: 10 ml/r;

rated pressure: 31.5 MPa; 7 pistons;

rated speed: 1500 rpm

3

Data acquisition card

USB-6221

Maximum sampling rate: 250 kS/s

4

Vibration sensor

YD-72D

Frequency range: 0.3 Hz–18 kHz;

Charge sensitivity: 0.35pC/(m/s2);

Linear range of amplitude: 1000m/s2.

5

Charge amplifier

DHF-10

Maximum output voltage: ±10V;

Gain: 0.1mV/pC–1V/pC; Precision: <1.5%.

6

Pressure Sensor

SYB-351

Measuring range: 0–25 MPa;

Precision: 0.2%; Output range: 0–5V.

7

Rotational speed measurer

LT-XSMP

Measuring range: 6–45000 rpm

Point 3: Figure 5. Physical diagram of the test system - is illegible, and rather Figure 5. Installation of an experimental installation.

Response 3: “Figure 5. Physical diagram of the test system” has been changed to “Figure 2. Installation of an experimental installation”.

Point 4: All cited sources concern only to Introduction part, none are assigned to the discussion or comparison of the results. Add more references with similar issues and focus of journal Sensor (which are totally lacking).

Response 4: Aiming at the problem that none cited sources are assigned to the discussion or comparison of the results, the fault diagnosis method in this paper is compared with the other two fault diagnosis methods, as shown in the “Response 1”.

In order to prove that the advantages of the fault diagnosis method proposed in this paper, hydraulic pump fault diagnosis was carried out using these methods under the same conditions. The diagnosis accuracies of the fault diagnosis methods [40-41] are shown in Table 7. Therefore, the advantages of the fault diagnosis method proposed in this paper, based on the integration of MEEMD, AR spectrum, and WKELM, were proved.

Table 7. Comparison of diagnostic accuracy of fault diagnosis methods.

Fault diagnosis method

MEEMD–WKELM

IMSE-KFCM[40]

RQA-KFCM[41]

Recognition accuracy

100%

98.75%

98%

40    Dong K. Improved Multi-scale Entropy and Apply to Fault Feature Extraction and Diagnosis of Rotating Machinery. Qinhuangdao: Yanshan University, 2016, 60-67.

41    Jiang, W.; Li, Z.; Zhang, S.; Lei, Y.; Wang, H. Fault Recognition Method Based on Recurrence Quantitation Analysis for Hydraulic Pump. Chinese Hydraulics & Pneumatics. 2019, (2), 18-23.

Some references with similar issues and focus of journal Sensor are added:

  1. Li, C.; Sánchez, RV.; Zurita, G.; Cerrada, M.; Cabrera, D. Fault Diagnosis for Rotating Machinery Using Vibration Measurement Deep Statistical Feature Learning. Sensors, 2016, 16(6), 895, doi: 10.3390/s16060895.
  2. Liu, T.; Luo, Z.; Huang, J.; Yan, S. A Comparative Study of Four Kinds of Adaptive Decomposition Algorithms and Their Applications. Sensors. 2018, 18(7), 2120, doi:10.3390/s18072120.
  3. Ahn, J.; Kwak, D.; Koh, B. Fault Detection of a Roller-Bearing System through the EMD of a Wavelet Denoised Signal. Sensors, 2014, 14(8), 15022-15038, doi: 10.3390/s140815022.
  4. Zhang, Y.; Liu, Y.; Chao, H.; Zhang, Z.; Zhang, Z. Classification of Incomplete Data Based on Evidence Theory and an Extreme Learning Machine in Wireless Sensor Networks. Sensors, 2018, 18(4), 1046, doi: 10.3390/s18041046.
  5. Cho, D.; Ham, J.; Oh, J.; Park, J.; Kim, S.; Lee, N.; Lee, B. Detection of Stress Levels from Biosignals Measured in Virtual Reality Environments Using a Kernel-Based Extreme Learning Machine. Sensors. 2017, 17(10), 2435. doi: 10.3390/s17102435.
  6. Zhang, S.; Zhang, T.; Yin, Y.; Xiao, W. Alumina Concentration Detection Based on the Kernel Extreme Learning Machine. Sensors, 2017, 17(9), 2002. doi: 10.3390/s17092002.
  7. Kuai, M.; Cheng, G.; Pang, Y.; Li, Y. Research of Planetary Gear Fault Diagnosis Based on Permutation Entropy of CEEMDAN and ANFIS. Sensors. 2018, 18(3), 782. doi: 10.3390/s18030782.
  8. Sun, J.; Xu, X.; Liu, Y.; Zhang, T.; Li, Y. FOG Random Drift Signal Denoising Based on the Improved AR Model and Modified Sage-Husa Adaptive Kalman Filter. Sensors. 2016, 16(7), 1073. doi: 10.3390/s16071073.
  9. Zhang, W.; Guo, W.; Zhang, C.; Zhao, S. An Online Calibration Method for a Galvanometric System Based on Wavelet Kernel ELM. Sensors. 2019, 19(6),1353. doi: 10.3390/s19061353.

Point 5: The results were not presented on the abstract. The authors needs to describe the main results in the abstract. Please correct.

Response 5: Aiming at the problem that the results were not presented on the abstract, the main results “The hydraulic pump fault diagnosis method presented in this paper can diagnose faults of single slipper wear fault, single slipper loosing fault and center spring worn fault with 100% accuracy, and the fault diagnosis time is only 0.002s.” has been added to the abstract.

Reviewer 2 Report

This manuscript proposes to solve the fault diagnosis problems using empirical mode decomposition and extreme learning machine. Several issues need to be solved before further consideration.

  1. The organization of the manuscript should be improved. Please simplify the preliminaries to emphasize the key components that formulate the problem. Besides, the discussions in the preliminaries, the proposed methods, and the experimental results seem to be messed up. Please reorganize Sections 2 to 4 to provide better readability.

  2. Section 2.4 can be moved to the experiments as it does not help to formulate the problem.

  3. The introduction part needs to be improved. Recent works in fault diagnosis are not extensively discussed. Papers you may refer: Unsupervised Cross-domain Fault Diagnosis Using Feature Representation Alignment Networks for Rotating Machinery.

  4. There are many types of modern neural networks, please explain why a single layer neural network is selected in this manuscript. Do other types of neural networks or deeper neural networks fit this problem as well? If so, please explain why they are not selected.

  5. The framework and detailed algorithm for the proposed method should be provided.

  6. Please provide the statistical test results for the fault diagnosis of the hydraulic pump.

  7. Please indicate how many rounds of trials were conducted? Please also report their variances.
  8. Please validate the proposed method on multiple datasets.

Author Response

Response to Reviewer 2 Comments

 You can also see the attachment.

Thank you for the reviewers' comments concerning our manuscript entitled "Hydraulic pump fault diagnosis method based on MEEMD and WKELM" (sensors-1160355). Those comments are all valuable and very helpful for revising and improving our paper, as well as the important guiding significance to our researches. We have studied comments carefully and have made correction which we hope meet with approval. Revised portion are marked in yellow in the paper. The main corrections in the paper and the responds to the reviewer's comments are as flowing:

Point 1: The organization of the manuscript should be improved. Please simplify the preliminaries to emphasize the key components that formulate the problem. Besides, the discussions in the preliminaries, the proposed methods, and the experimental results seem to be messed up. Please reorganize Sections 2 to 4 to provide better readability.

Response 1: We have simplified the preliminaries,including the deletion of the fourth paragraph of the Sections 1:“Adaptive analysis methods mainly include Empirical Mode Decomposition (EMD) and its improved methods. EMD is a time–frequency signal processing method, which was proposed by Huang in 1998 [11]. EMD method can decompose non-linear and non-stationary signals into multiple Intrinsic Mode Functions (IMFs), which can fully reflect the local characteristics of the original signal. To date, the EMD method has been widely used in various fields [12]. However, the EMD decomposition of non-linear and non-stationary signals often produces the phenomenon of end-effect and mode mixing, which makes the decomposed IMF components quite confusing.”and simplified the Sections 1:Considering the mode mixing phenomenon of the EMD method, Wu et al. proposed the Ensemble Empirical Mode Decomposition (EEMD) method, which effectively suppresses modal mixing by adding Gaussian white noise to the original signal [10]. However, the white noise added to the original signal cannot be completely eliminated, such that more false modal components will be decomposed. In view of the shortcomings of the EEMD method, scholars have carried out a lot of research. Finally, Zheng Jinde et al. proposed the Modified Ensemble Empirical Mode Decomposition (MEEMD) method [11], which adds a pair of Gaussian white noises with the same amplitude and opposite signs, such that they cancel each other and the influence of the noise on the signal is reduced. Compared with the EEMD method, this method has a better decomposition effect. Jiang Shuangyang applied MEEMD and a Cloud Particle Swarm Optimization (CPSO)-based Support Vector Machine (SVM) method to the fault diagnosis of the main shaft bearing of a wind turbine [12]. Their method showed good results, in terms of training accuracy and training speed, which verifies the feasibility of this method. Zheng Xu proposed a method combining MEEMD and Generalized Adaptive S Transform (AGST), which was successfully applied to the vibration signal analysis of an internal combustion engine [13]. Shi Yuancheng et al. applied the MEEMD method to fault feature extraction of rolling bearings, which not only effectively suppressed the phenomenon of modal mixing but also greatly reduced the number of pseudo-components and accurately extracted the weak fault information of rolling bearings [14]. Although the MEEMD method has been widely used in the field of fault diagnosis, it has seldom been reported in hydraulic system fault diagnosis. Therefore, in this paper, we applied it to the vibration signal feature extraction of an axial piston pump, in order to improve the accuracy of fault identification.

In 2004, Professor Huang from Nanyang Technological University in Singapore proposed a new single-hidden layer feedforward neural network, called the Extreme Learning Machine (ELM) [15]. The key characteristic of this method is that the input layer weights and hidden layer thresholds can be selected randomly, and the two parameters do not need to be adjusted iteratively during the process of training. The number of hidden neurons just needs to be set before training. In order to optimize the number of hidden layer neurons, Huang et al. proposed an Incremental Extreme Learning Machine (I-ELM) in 2006. This method uses the method of incremental construction to iterate the number of hidden layer neurons until the training accuracy reaches a satisfactory value, in order to obtain the optimal number of hidden layer neurons [16]. However, I-ELM has the disadvantage of long training time. In 2007, Huang et al. introduced a convex optimization learning method for the I-ELM, which greatly reduced its training time [17]. In 2013, Li et al. proposed a method combining I-ELM and Schmidt orthogonalization, which effectively improved the training accuracy [18]. However, it should be noted that the optimal number of neurons in the hidden layer obtained by the incremental extreme learning machine and its improved method may be very large, thus wasting a lot of unnecessary training time.

We also reorganized Sections 2 to 4, including exchanged Sections 2.1 and Sections 2.2, moved Section 2.4 to the experiments, moved Section 4.1 to Section 4.3.

Point 2: Section 2.4 can be moved to the experiments as it does not help to formulate the problem.

Response 2: We have moved the Section 2.4 to the experiments.

Point 3: The introduction part needs to be improved. Recent works in fault diagnosis are not extensively discussed. Papers you may refer: Unsupervised Cross-domain Fault Diagnosis Using Feature Representation Alignment Networks for Rotating Machinery.

Response 3: For this problem, we retrieved the latest fault diagnosis literatures and added some references, as follows:

  1. Li, C.; Sánchez, RV.; Zurita, G.; Cerrada, M.; Cabrera, D. Fault Diagnosis for Rotating Machinery Using Vibration Measurement Deep Statistical Feature Learning. Sensors, 2016, 16(6), 895, doi: 10.3390/s16060895.
  2. Liu, T.; Luo, Z.; Huang, J.; Yan, S. A Comparative Study of Four Kinds of Adaptive Decomposition Algorithms and Their Applications. Sensors. 2018, 18(7), 2120, doi:10.3390/s18072120.
  3. Chen, J.; Wang, J.; Zhu, J.; Lee, T.; Silva, C. Unsupervised Cross-domain Fault Diagnosis Using Feature Representation Alignment Networks for Rotating Machinery. IEEE/ASME Transactions on Mechatronics. 2020, 99, 1-1, doi: 10.1109/TMECH. 2020.3046277.
  4. Chen, J.; Li, T.; Wang, J.; Silva, C. WSN Sampling Optimization for Signal Reconstruction Using Spatiotemporal Autoencoder. IEEE Sensors Journal. 2020, 99, 1-1, doi: 10.1109/JSEN.2020.3007369.
  5. Chen, J.; Li, T.; Wang, J.; Silva, C. Optimization of Wireless Sensor Network Deployment for Spatiotemporal Reconstruction and Prediction. Electrical Engineering and Systems Science. 2019.
  6. Ahn, J.; Kwak, D.; Koh, B. Fault Detection of a Roller-Bearing System through the EMD of a Wavelet Denoised Signal. Sensors, 2014, 14(8), 15022-15038, doi: 10.3390/s140815022.
  7. Zhang, Y.; Liu, Y.; Chao, H.; Zhang, Z.; Zhang, Z. Classification of Incomplete Data Based on Evidence Theory and an Extreme Learning Machine in Wireless Sensor Networks. Sensors, 2018, 18(4), 1046, doi: 10.3390/s18041046.
  8. Cho, D.; Ham, J.; Oh, J.; Park, J.; Kim, S.; Lee, N.; Lee, B. Detection of Stress Levels from Biosignals Measured in Virtual Reality Environments Using a Kernel-Based Extreme Learning Machine. Sensors. 2017, 17(10), 2435. doi: 10.3390/s17102435.
  9. Zhang, S.; Zhang, T.; Yin, Y.; Xiao, W. Alumina Concentration Detection Based on the Kernel Extreme Learning Machine. Sensors, 2017, 17(9), 2002. doi: 10.3390/s17092002.
  10. Kuai, M.; Cheng, G.; Pang, Y.; Li, Y. Research of Planetary Gear Fault Diagnosis Based on Permutation Entropy of CEEMDAN and ANFIS. Sensors. 2018, 18(3), 782. doi: 10.3390/s18030782.
  11. Sun, J.; Xu, X.; Liu, Y.; Zhang, T.; Li, Y. FOG Random Drift Signal Denoising Based on the Improved AR Model and Modified Sage-Husa Adaptive Kalman Filter. Sensors. 2016, 16(7), 1073. doi: 10.3390/s16071073.
  12. Zhang, W.; Guo, W.; Zhang, C.; Zhao, S. An Online Calibration Method for a Galvanometric System Based on Wavelet Kernel ELM. Sensors. 2019, 19(6),1353. doi: 10.3390/s19061353.
  13. Chen, J.; Shu, T.; Li, T.; Silva, C. Deep Reinforced Learning Tree for Spatiotemporal Monitoring With Mobile Robotic Wireless Sensor Networks. IEEE Transactions on Systems, Man, and Cybernetics: Systems. 2019, 1-15, doi: 10.1109/TSMC.2019.2920390.

For the problem that recent works in fault diagnosis are not extensively discussed, we added some references:

In order to prove that the advantages of the fault diagnosis method proposed in this paper, hydraulic pump fault diagnosis was carried out using these methods under the same conditions. The diagnosis accuracies of the fault diagnosis methods [40-41] are shown in Table 8. Therefore, the advantages of the fault diagnosis method proposed in this paper, based on the integration of MEEMD, AR spectrum, and WKELM, were proved.

Table 8. Comparison of diagnostic accuracy of fault diagnosis methods.

Fault diagnosis method

MEEMD–WKELM

IMSE-KFCM[40]

RQA-KFCM[41]

Recognition accuracy

100%

98.75%

98%

  1. Dong K. Improved Multi-scale Entropy and Apply to Fault Feature Extraction and Diagnosis of Rotating Machinery. Qinhuangdao: Yanshan University, 2016, 60-67.
  2. Jiang, W.; Li, Z.; Zhang, S.; Lei, Y.; Wang, H. Fault Recognition Method Based on Recurrence Quantitation Analysis for Hy-draulic Pump. Chinese Hydraulics & Pneumatics. 2019, (2), 18-23.

Point 4: There are many types of modern neural networks, please explain why a single layer neural network is selected in this manuscript. Do other types of neural networks or deeper neural networks fit this problem as well? If so, please explain why they are not selected.

Response 4: WKELM is single-hidden layer feedforward neural network and the method proposed in this paper has achieved a high accuracy. We have tried to use deep neural network, but the fault identification accuracy could not be effectively improved, and the diagnosis time was long, so it was not suitable for real-time fault diagnosis. We also tried to use other single-layer neural networks have been used, the calculation speed is slow and the accuracy is also low, as shown in Table 6. If other deep neural networks are used, it is easy to fall into overfitting and the network structure is complex, which has little effect on improving the accuracy, so other types of neural networks or deeper neural networks does not fit this problem.

Point 5: The framework and detailed algorithm for the proposed method should be provided.

Response 5: The framework and flow chart of the proposed method is shown in 14. The detailed algorithm for the proposed method has been provided at Section 2 and Section 3.

Point 6: Please provide the statistical test results for the fault diagnosis of the hydraulic pump.

Response 6: Statistical test results as shown in table 5, the experiment was carried out in four groups, including normal working state, single slipper wear fault state, single slipper loosing fault state, center spring worn fault state, each group of tests has done 2 times, each experiment consisted of 50 samples. Then randomly selected 30 samples as training samples, the remaining 20 as the test sample. Therefore, the training sample set 30*4=120 samples and test sample sets 20*4=80 samples, the test to identify the results as shown in table 5, the results of each fault diagnosis method was tested five times and averaged.

Table 5. Comparison of diagnostic accuracy of four fault diagnosis methods.

Condition of hydraulic pump

Normal working

Single slipper wear fault

Single slipper loosing fault

Center spring worn fault

Total

MEEMD–BP

Number of samples correctly diagnosed

20

20

20

19

79

Recognition accuracy

100%

100%

100%

95%

98.75%

MEEMD–SVM

Number of samples correctly diagnosed

20

19

19

20

78

Recognition accuracy

100%

95%

95%

100%

97.5%

MEEMD–ELM

Number of samples correctly diagnosed

20

19

20

20

79

Recognition accuracy

100%

95%

100%

100%

98.75%

MEEMD–WKELM

Number of samples correctly diagnosed

20

20

20

20

80

Recognition accuracy

100%

100%

100%

100%

100%

Point 7: Please indicate how many rounds of trials were conducted? Please also report their variances.

Response 7: The experiment was carried out in four groups, including normal working state, single slipper wear fault state, single slipper loosing fault state, center spring worn fault state, each group of tests has done two times. Each group of tests was carried out two times, and data were collected five times for each test, and each acquisition time was 1s. In each test, the piston pump's failure mode was changed by replacing the normal components with fault components. And then, the intrinsic mode of the vibration signal will be changed. For the problem that the variance of many rounds of trials did not reported in this paper, the method proposed in this paper has been tested for many rounds of trials and the recognition accuracy is 100%, so the variance is 0.

Point 8: Please validate the proposed method on multiple datasets.

Response 8: The Case Western Reserve University data was used to demonstrate the fault diagnosis method proposed in this paper has high fault recognition accuracy. The sampling frequency of the data is 12kHz, the model of the bearing is SKF6205, the damage diameter is 0.36mm, the motor loads is 2 horsepower. Take the normal working state, rolling ball fault state, inner race fault state, and outer race fault data, 0.2 s for each state data as a sample, take 50 samples under each state, of which 30 as the training sample, 20 as test samples, the results as shown in table 7. It can be seen that the fault diagnosis method proposed in this paper has a high recognition accuracy.

Table 7. Fault diagnosis of rolling bearings based on WKELM.

Condition of rolling bearing

Normal working

Inner race fault

rolling ball fault

Outer race fault

Total

Number of samples correctly diagnosed

20

20

20

20

80

Recognition accuracy

100%

100%

100%

100%

100%

Round 2

Reviewer 2 Report

The authors have addressed my previous comments, and the quality of the manuscript has been improved significantly.